# Gut Microbiome: A Promising Biomarker for Immunotherapy in Colorectal Cancer

**DOI:** 10.3390/ijms20174155

**Published:** 2019-08-25

**Authors:** Sally Temraz, Farah Nassar, Rihab Nasr, Maya Charafeddine, Deborah Mukherji, Ali Shamseddine

**Affiliations:** Department of Internal Medicine, Hematology/Oncology Division, American University of Beirut Medical Center, Riad El Solh, Beirut 1107 2020, Lebanon

**Keywords:** colorectal cancer, gut microbiome, immunotherapy

## Abstract

Research has been driven towards finding therapy predictive biomarkers for colorectal cancer (CRC) with a special interest in studying the gut microbiome. Gut microbiome acts not only as a barrier to prevent bacterial invasion and infection, but it also affects the efficacy of hematopoietic-cell transplantation, chemotherapy, and immunotherapy. Recently, immunotherapy, which potentiates the host immune system, has revolutionized cancer therapy in general and CRC treatment specifically by increasing the quality of life and the survival of a subset of patients with this disease. In immunotherapy, the gut microbiome plays an important role in cytotoxic T-lymphocyte-associated antigen 4 (CTLA-4) blockade, programmed cell death protein 1 (PD-L1) mediation, and T cell stimulation. As such, this review will cover the role of gut microbiome in CRC, summarize approved immunotherapy treatments for CRC, and focus on the potential use of gut microbiome as a biomarker for immunotherapy.

## 1. Introduction

Colorectal cancer (CRC) is the third most common cancer type worldwide (10.2%). Despite early screening and new treatment strategies, CRC remains the second most fatal cancer worldwide (9.2%) [1], with increased mortality rates in countries with more inadequate resources and health infrastructure, especially in Central and South America and Eastern Europe [2]. Early stage CRC (stage I–III) patients usually have a good prognosis with a five-year survival exceeding 50%; however, patients with metastatic disease have a five-year survival rate of less than 12% [3]. Research has been driven towards finding therapy predictive biomarkers for CRC with a special interest in studying the gut microbiome. Recently, immunotherapy, which utilizes the host immune system, has revolutionized cancer therapy in general and CRC treatment specifically by increasing the quality of life and the survival of a subset of patients. As such, this review will cover the role of gut microbiome in CRC, summarize approved immunotherapy treatments for CRC, and focus on the potential use of gut microbiome as a biomarker for immunotherapy.

### 1.1. An Overview of the Gut Microbiome

The human microbiota is composed of 10–100 trillion microbes including bacteria, viruses, protozoa, and fungi, with the largest population residing in the gut reaching around 1011–1012 cells/mL. The microbiota harbored in the gut has a biomass of 1.5 kg and its composition is influenced by age, diet, environment, gender, ethinicity, and the co-evolutionary forces between microbial communities and their hosts [4,5,6,7]. The predominant phyla of gut bacteria are *Firmicutes* and *Bacteroidetes* [8]. The gut microbiota is considered a separate organ playing a vital role in the human health; it helps in food digestion and metabolism from otherwise indigestible compounds [9,10], produces essential vitamins (B and K) [11], acts as a barrier to pathogens [12], and stimulates the gut immune system [13]. Furthermore, the development of metagenomic sequencing technology and analysis has revealed that the dysbiosis or imbalance in the normal intestinal microbiota can act as a risk factor to several disorders including allergies [14]; obesity; diabetes [15]; inflammatory bowel disease [16]; irritable bowel syndrome [17]; and many types of cancer, especially CRC [18,19,20].

### 1.2. Colorectal Cancer and Gut Microbiome

The first correlation of gut microbiota with CRC was reported in 1975 upon observing an increased development of colon adenomas in germ-free rats as compared with conventional rats [21]. It is now established that CRC patients have distinct microbiota compared with healthy subjects [22]. Two models are present to explain the contribution of gut microbiome in CRC pathogenesis. In the first model, gut microbiota act as “drivers” with pro-carcinogenic features that may initiate CRC development by inducing epithelial DNA damage, and which then could be replaced by “passenger” bacteria that can promote or hinder carcinogenesis, and have a growth advantage in the tumor microenvironment [23]. The other model takes into consideration the host genetics, which allow the dysbiosis of microbial community as a whole, causing pro-inflammatory responses and epithelial cell transformation and eventually leading to cancer. Several studies in CRC patients and in experimental animals have linked dysbiosis of the gut microbial composition of the tumor and adjacent mucosa with CRC [24]. Dysbiosis of the gut microbiome includes the expansion and depletion of certain bacterial species. Specific enteric virome signatures were also identified in fecal samples from CRC patients as compared with controls. Some viral markers were associated with reduced survival of CRC patients [25]. *Fusobacterium nucleatum*, a gram negative anaerobe, is the most prevalent gut bacterium in CRC [26,27] and is validated in different ethnic groups [28,29,30,31]. Antibiotic administration by mice bearing CRC xenograft caused not only a decrease in *Fusobacterium* load, but also a reduction in cancer cell proliferation and overall tumor growth [32]. *Fusobacterium nucleatum* has been suggested as a prognostic biomarker in CRC, as its high levels in CRC tissues significantly correlated with shorter overall survival [33,34]. Variations in the studies about CRC microbiota are reported, and are mainly attributed to differences in the nature of the sampling (feces vs. mucosal tissue) or differences in stages or location of the disease [22,35].

As such, the gut microbiome is currently investigated as a potential biomarker for CRC early detection and prognosis. A recent study profiled the fecal microbiomes in 74 CRC patients and 54 controls from China through metagenomics sequencing and validated the results in ethnically different cohorts. This study found significant enrichment of novel species, including *Parvimonas micra* and *Solobacterium moorei*, along with already established association of *Fusobacterium nucleatum* and *Peptostreptococcus stomatis*, with CRC. It also highlighted the potential use of microbial gene markers as fecal metagenomic biomarkers for early diagnosis of CRC [36].

### 1.3. Immunotherapy and Colorectal Cancer

Because CRC is a heterogeneous disease, genetic analysis of the tumor is increasingly performed to decide whether biological agents such as immunotherapy are to be used. Currently, immunotherapy is approved for only a small subset of patients whose tumor is characterized by high microsatellite instability (MSI-H) phenotype. Meanwhile, surgical resection is usually the treatment option for stages I–III of localized CRC, while chemotherapy and especially fluorouracil based treatment is usually the standard therapy for metastatic CRC.

Microsatellites are repeated DNA sequences of 2–5 base pairs in length and are scattered throughout the genome. Their high instability (MSI-H) refers to the high rate of somatic mutations accumulating in their region that result from an impaired DNA mismatch repair (MMR), which is a system responsible for recognition and repair of DNA damage. Deficient MMR (dMMR) can occur either by germline mutations of genes encoding for MLH 1, MSH 2, MSH 6, and PMS2 protein as in Lynch syndrome (also called hereditary nonpolyposis colorectal cancer (HNPCC)) or by sporadic hypermethylation of the MLH 1 gene promoter [37]. CRC patients with hereditary or sporadic MSI-H/dMMR tend to have the same prognosis [37]. The frequency of MSI-H is 15–20% in overall CRC patients and 5% in metastatic CRC [38]. Data have shown that identifying tumors as MSI-H acts as an effective predictive biomarker for CRC treatment management, particularly oxaliplatin or irinotecan based therapy and recently immunotherapy [39,40,41].

Currently, three U.S. Food and Drug Administration (FDA)-approved immunotherapy options are present for metastatic MSI-H CRC patients. Immunotherapy targets immune checkpoints molecules required for activating the immune system such as cytotoxic T-lymphocyte-associated antigen 4 (CTLA-4), programmed cell death protein 1 (PD-1), and programmed death-ligand 1 (PD-L1). In mid 2017, pembrolizumab (Keytruda), a humanized monoclonal antibody that blocks the interaction between PD-1 and its ligands, got accelerated FDA approval for treating either MSI-H or dMMR CRC that progressed on combination treatment of fluoropyrimidine; oxaliplatin; and irinotecan or any unresectable or metastatic, MSI-H, or dMMR solid tumors that progressed on a previous treatment and have no other treatment options [42]. This approval was granted after five multi-centered clinical trials on 149 cancer patients; 90 of whom had CRC. According to RECIST 1.1, 11 patients had complete responses and 48 had partial responses. The objective response rate (ORR) was 39.6%, which was similar between those with CRC (36%) or other cancer type (46%).

Nivolumab (Opdivo), a PD-1 immune checkpoint inhibitor, was also granted accelerated FDA approval in August 2017 for patients with MSI-H or dMMR metastatic CRC that have progressed after chemotherapy. In a phase II clinical trial (CheckMate 142), MSI-H or dMMR metastatic CRC patients were given Nivolumab monotherapy and were monitored for 12 months. After this period, all participants remained alive and 23 out of 74 patients achieved an investigator-assessed objective response and 51 patients had disease control for 12 weeks or longer [43]. The latest immunotherapy combination that was approved by the FDA in July 2018 is Nivolumab (Opdivo) and ipilimumab (Yervoy) for MSI-H CRC. Ipilimumab, a monoclonal antibody that targets the CTLA-4 checkpoint receptor, was shown to act synergistically with Nivolumab to cause T-cell antitumor activity in melanoma and small lung cell carcinoma [44,45]. In a large phase 2 clinical trial (CheckMate-142) on 119 MSI-H CRC patients, Nivolumab plus ipilimumab demonstrated higher response rates (55%), progression-free survival (71%), and OS (85%) at 12 months, and significant improvement in patient-reported outcomes including functioning, symptoms, and quality of life as compared with Nivolumab monotherapy [46]. Even though this trial mainly included patients who were intolerant or progressed following fluoropyrimidine and oxaliplatin or irinotecan-containing regimens, it suggests that nivolumab and ipilimumab could be used as a first-line treatment option for MSI-H CRC patients.

## 2. Role of Gut Microbiome in Modulating Immune Response to Cancer Treatments

The gut microbiome has an essential role in the development and the regulation of adaptive and innate immunity (Figure 1). This role of gut microbiome is evident in germ-free mice that reside in an environment devoid of microorganisms when compared with normal mice. Germ-free mice develop a defective immune system, especially in the gut, with an altered mucosal layer; a decrease in size and functionality of Peyer’s patches and lymphoid tissues; as well as reduction in immune cell counts, microbe sensing toll-like receptors (TLR), and major histocompatibility complex II molecules for antigen presentation [47,48,49,50]. The gut microbiome acts not only as a barrier to prevent bacterial invasion and infection, but it also affects the efficacy of hematopoietic-cell transplantation and chemotherapy.

### 2.1. Hematopoietic-Cell Transplant

A prospective study on 541 patients admitted for hematopoietic-cell transplantation revealed an association between the abundance of a bacterial group, mostly *Eubacterium limosum*, and the relapse/progression of hematological malignances after allogenic hematopoietic-cell transplantation [51].

### 2.2. Chemotherapy

In mouse models, the use of cyclophosphamide aided the translocation of gram positive bacteria into secondary lymphoid organs, where they stimulated the generation of “pathogenic” T helper 17 cells and memory Th1 immune responses. Meanwhile, resistance to cyclophosphamide was noted in mice that had reduced T helper 17 responses upon being germ-free or treated with antibiotics [52]. This was later confirmed by another study where two bugs, *Enterococcus hirae* and *Barnesiella intestinihominis*, were found to facilitate the anticancer effect of cyclophosphamide in lung and ovarian cancers by increasing the intratumoral CD8/Treg ratio in secondary lymphoid organs and by infiltrating IFN-γ-producing γδT cells into cancer lesions, respectively [53]. In another study that highlights the importance of commensal bacteria in modulating tumor microenvironment, subcutaneous EL4 lymphoma tumor-bearing mice that were germ-free or treated with antibiotics had an impaired response to oxaliplatin as a result of deficiency in reactive oxygen species production and cytotoxicity.

### 2.3. Immunotherapy

MC38 colon adenocarcinoma tumor-bearing mice that were germ-free or treated with antibiotics had a defective response to CpG-oligonucleotide immunotherapy treatment as a result of low cytokine production and tumor necrosis [54]. As such, it has been suggested that the gut microbiome can modulate the immune system and affect the efficacy of immunotherapy (Figure 1). The gut microbiota was found to be vital for the antitumor effect of blocking CTLA-4, particularly through the T cell responses of *Bacteroidales* (*B. fragilis* and/or *B. thetaiotaomicron*) and *Burkholderiales*. It was demonstrated at first in vivo in germ-free and antibiotic treated mice that were transplanted with colon and melanoma tumors and did not respond to CTLA-4 blockade. This defect in response to immunotherapy was overcome when the mice orally ingested *B. fragilis*, when mice were immunized with *B. fragilis* polysaccharides, or when they had adoptive transfer of *B. fragilis*-specific T cells. Furthermore, the clinical relevance was highlighted when an outgrowth of *B. fragilis* with anticancer properties occurred in germ-free mice that were transplanted with feces harvested from 25 different metastatic melanoma patients treated with ipilimumab, CTLA-4 blocker [55].

The gut microbiota also plays an important role in mediating PD-L1 efficacy (Figure 1). Commensal *Bifidobacterium* was associated with a clinical benefit for PD-L1 checkpoint blockade. Differences in response to anti-PD-L1 therapy and in melanoma growth and aggressiveness were noted in genetically similar C57BL/6 mice derived from two facilities, Jackson Laboratory (JAX) and Taconic Farms (TAC). The difference in response to anti-PD-L1 therapy was decreased when TAC orally administrated JAX fecal material. This was mainly attributed to the commensal bacteria *Bifidobacterium* that stimulates dendritic cell maturation and eventually increases the effector function of tumor-specific CD8+ T cells. [56]. Furthermore, antibiotic-caused dysbiosis of gut microbiome was correlated with unresponsiveness to immunotherapy. Antibiotic administration by patients with non-small-cell lung carcinoma, renal cell carcinoma, and urothelial carcinoma decreased the response to PD-1 blockade and shortened survival compared with patients who did not use antibiotics. Higher abundance of *Akkermansia muciniphila* (*A. muciniphila*) was identified in fecal microbiota of responders to PD-1 blockade and its oral supplementation from responders in epithelial tumors of murine models restored PD-1 blockade. *A. muciniphila* affects the efficacy of immunotherapy as it was shown to induce IL-12 secretion by dendritic cells, which leads to the recruitment of CCR9+CXCR3+CD4+ T lymphocyte cells into mouse tumor [57]. Another research group did a similar study on the difference of gut microbiome in metastatic melanoma patients. Responders to anti-PD-*Bacteroidales* 1 immunotherapy had higher abundance of *Ruminococcaceae* and *Faecalibacterium*, which increased antigen presentation and improved effector T cell function in the periphery and the tumor microenvironment, while non-responders had higher abundance of *Bacteroidales*, which mediated limited intratumoral lymphoid and myeloid infiltration and weakened antigen presentation capacity [58]. Similarly, Matson et al. studied the effect of gut microbiome of metastatic melanoma patients on their response to immunotherapy using a different method in analysis and in patient segregation between responder and non-responder groups [59]. Their group found, after sequencing stool samples before immunotherapy, an increased abundance of eight microbial species, including *Bifidobacterium longum*, *Collinsella aerofaciens*, and *Enterococcus faecium* in responders. Interestingly, two species were also associated with non-responsiveness (*Ruminococcus obeum* and *Roseburia intestinalis*). Colonization of germ-free, tumor-bearing mice with feces from responder patients showed improved tumor control and better efficacy to anti-PD-L1 therapy.

A recent paper revealed that Rnf5−/− melanoma bearing mice that lack ring finger protein 5 (RNF5) (helps cells remove incorrectly folded proteins) and are characterized by low unfolded protein response (UPR) activity benefit from immunotherapy, because they have altered gut microbiome contributing to antitumor immunity [60]. Reduced UPR component expression (sXBP1, ATF4, and BiP) was reported not only in mouse melanoma tumors that responded to CTLA4 therapy, but also in patients who responded to immune checkpoint therapy. Eleven strains of gut bacteria, including *B. rodentium*, enriched in Rnf5−/− mice, slow the growth of melanoma tumors (Figure 1).

## 3. Gut Microbiome Role in CRC Response to Immunotherapy

The above-mentioned studies focused on the role of enteric microorganisms in the treatment of melanoma by immune checkpoint inhibitors, but not CRC. Few studies have focused on the role of immune checkpoint inhibitors in CRC, as the percentage of those patients is low. However, the results from melanoma studies give a glimpse on how gut microbe can have a similar role in CRC response to immunotherapy. A recent study by Tanoue et al. distinguished 11 healthy human associated bacterial strains that act together to induce interferon γ CD8 Tcells, confer resistance to the intracellular pathogen Listeria, and are effective at inhibiting tumor growth when used with immune checkpoint inhibitors [61]. Besides, the gut microbiome was found to stimulate T cell trafficking into CRC tumor tissues [62]. Expression of T cell-recruiting chemokines was significantly correlated with an increase in *E. coli* and *B. fragilis* in vitro and with abundance of *Firmicutes*, in particular *Lachnospiraceae* and *Ruminococcaceae*, as well as *Bacteroides* and *Proteobacteria* ex vivo (Figure 1). Toll like receptors (TLR) on CRC cells were suggested to be not only the mediator for sensing gut micro-organisms, but also the inducer of increased chemokine gene expression. This was observed when adding TLR agonist to CRC cells. Interestingly, *Fusobacteria*, which is the main bacteria correlated with poor prognosis, was associated with increased expression of T cell recruiting chemokine genes only in vitro. This suggested that other bacteria can act as attractant for immune cells and *Fusobacteria* might have another function. For example, *Fusobacteria* was shown to have a role in tumor evasion because it can inhibit natural killer cells and tumor-infiltrating lymphocytes via binding its Fap2 protein to human inhibitory receptor TIGT [63]. On the other hand, a study found that *Fusobacterium nucleatum* secretes interleukin (IL)-12 and transforming growth factor (TGF)-β, which promotes differentiation of nonsuppressive T lymphocytes that have low FOXP3 expression and predictive of favorable survival [64]. *F. nucleatum* was also recently shown to increase the expression of inflammatory mediators such as IL1B, IL6, and IL8 through a possible miRNA-mediated activation of TLR2/TLR4 [65] (Figure 2).

Notably, *Fusobacterium* in CRC was shown to be associated with lower-level T-cell infiltrates, poor clinical outcomes, and microsatellite instability (MSI), independent of CpG island methylator phenotype (CIMP) and BRAF mutation status [28,66,67,68]. *F. nucleatum* was suggested to interact with immune response differentially by tumor microsatellite instability. *F. nucleatum* was negatively correlated with tumor-infiltrating lymphocytes (TIL) in MSI-high tumors, but positively associated with TIL in non-MSI-high tumors [69]. RNA sequencing of 34 CRC tissue samples showed distinct bacterial communities associated with each CRC molecular subtype divided according to four consensus molecular subtypes annotated by Guinney et al. [70,71]. Enrichment of *Fusobacteria* and *Bacteroidetes*, and reduced levels of *Firmicutes* and *Proteobacteria*, were reported in the most aggressive subtype CMS1 (consensus molecular subtype 1), that is, MSI-H, CIMP high, hypermutated, and with BRAF mutation. Another study that sequenced paired colon tumor and normal-adjacent tissue and mucosa samples revealed significant enrichment of *Bacteroides fragilis* and *Fusobacterium nucleatum* in dMMR CRC, but not in proficient MMR CRC [72]. Interestingly, their results were also validated in metabolic modeling and metabolomics.

## 4. Conclusions

Even though many studies have highlighted the role of gut microbiome in response to immunotherapy in cancer, further investigations are still required to consider gut microbiome a predictive biomarker for immunotherapy response in CRC. First, characterization of the microbial composition is essential and optimal profiling methods (16S rRNA sequencing versus metagenomic shotgun sequencing and choice of reference databases) are yet to be identified. Moreover, the consensus molecular subtypes (CMS1, CMS2, and CMS3) ought to be taken into consideration as each subtype was associated with specific bacterial species [70], as well as tumor location in CRC (left or right sided). Second, several confounding factors should be taken into consideration when drawing conclusions from clinical studies, including patient’s diet, administered medications (especially probiotics and antibiotics), mental health, and environmental factors. Third, efficacy and resistance to immunotherapy should be identified in the changes of gut microbiome before and after administrating therapy, as it is not yet clear what composition of the gut microbiome is optimal to facilitate anti-tumor immune responses. Microbial intervention is being explored in conjunction with cancer therapeutics such as immunotherapy, chemotherapy, and hematopoietic stem cell transplantation in the form of fecal microbial transplantation, prebiotics, probiotics, live bacteria with phages, and dietary approaches. This is a promising therapy, but still requires much caution and fine tuning due to the diverse function of gut microbiome in the human body as a whole.

## Figures and Tables

**Figure 1 ijms-20-04155-f001:**
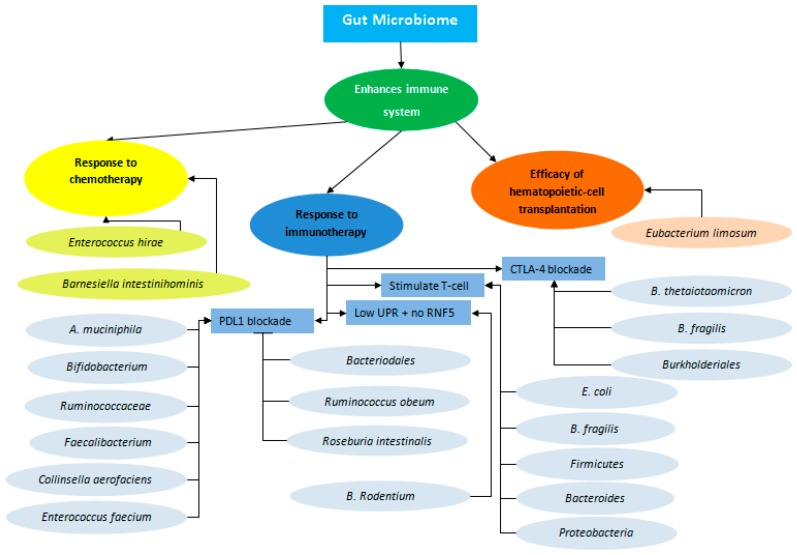
Gut microbiome plays an essential role in response to cancer treatment. Specific bacteria in the gut microbiome influence response to chemotherapy (shown in yellow ovals), hematopoietic stem cell transplantation (shown in orange ovals), and immunotherapy (shown in blue ovals). Bacterial strains influencing immunotherapy do so by either inhibiting or enhancing programmed cell death protein 1 (PDL1) blockade, by stimulating T cell, or by stimulating cytotoxic T-lymphocyte-associated antigen (CTLA) blockade. Melanoma cells with no ring finger protein 5 (RNF5) and low unfolded protein response (UPR), which were administered *B. Rodentium*, also improved response to immunotherapy.

**Figure 2 ijms-20-04155-f002:**
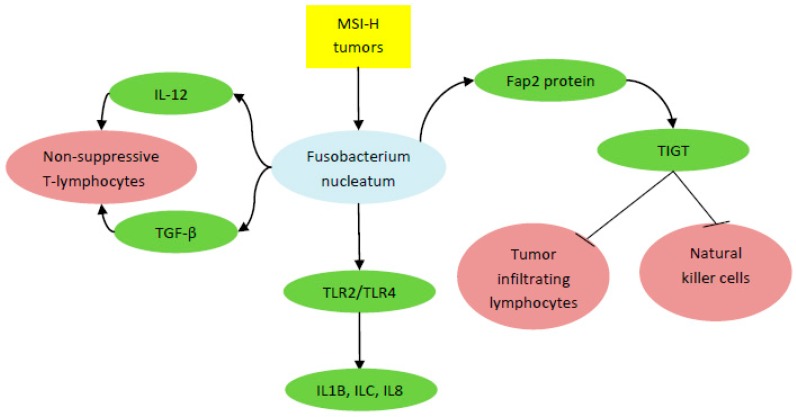
*Fusobacterium* correlated with poor outcome in high microsatellite instability (MSI-H) colorectal tumors by activating inflammatory mediators (interleukin (IL)1B, IL6, and IL8), secreting IL-12 and transforming growth factor β (TGF-β), and inhibiting natural killer cells and tumor infiltrating lymphocytes. TLR, Toll-like receptor.

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
