# Peer review of "Gut Microbiome: A Promising Biomarker for Immunotherapy in Colorectal Cancer"

_ijms, 2019, doi:10.3390/ijms20174155_

Round 1
Reviewer 1 Report
This is a good review in the area of microbiota and its importance in immunotherapy in cases of CRC. The manuscript is well written and cites most of the relevant references.
1) The english could be slightly polished. Colloquial phrases like "A lot" could replaced with more formal wording.
2) The words "In vitro" and "in vivo" should always be italicized (in vitro, in vivo)
3) The names of bacterial species: F. nucleatum or E. coli should always be italicized
4) The authors do not cite the work from Tanoue et al., Nature 2019 in which they identify a defined group of commensal bacteria that potentiate or prime the immune system for immunotherapy.
Author Response
This is a good review in the area of microbiota and its importance in immunotherapy in cases of CRC. The manuscript is well written and cites most of the relevant references.
1) The english could be slightly polished. Colloquial phrases like "A lot" could replaced with more formal wording.
We thank the reviewer and have replaced colloquial phrases with formal wording.
2) The words "In vitro" and "in vivo" should always be italicized (in vitro, in vivo)
Thank you we have italicized the words “in vitro” and “in vivo”.
3) The names of bacterial species: F. nucleatum or E. coli should always be italicized
We thank the reviewer for this comment and have italicized all bacteria names in the text
4) The authors do not cite the work from Tanoue et al., Nature 2019 in which they identify a defined group of commensal bacteria that potentiate or prime the immune system for immunotherapy.
We thank the reviewer for this important remark and have added the work from Tanoue et al (cited as reference 61 in the text).
Reviewer 2 Report
The authors have written a short review on the association between colorectal cancer treatment outcomes and microbiome, highlighting the role of the microbiome in immunotherapy.
1. Line 82-85: Expecting authors to talk about immunotherapy use. Suggest restructure paragraph so that immunotherapy comes first.
2. Line 128 Section 2: title is too vague. Consider renaming section: Role of gut microbiome in modulating immune response to cancer treatments
3. Figure 1: Ambitious and perhaps misleading. Currently diagram suggests that the gut microbiome directly affects response to immunotherapy but this happens via immune system. Described in text as role in development and regulation of adaptive immunity; Suggest remove beneficial effects and dysbiosis and focus on modification of response to therapy via immune system. Write more thorough description to accompany new figure.
4. Line 216: reword sentence to simplify
5. Line 243: the worst? Most aggressive?
6. Figure 2: Needs description
7. List of abbreviations not complete: e.g. dMMR , TGF, CMS1
8. Conclusion: Authors suggest 3 steps for further research. More detail is needed on each of these. For example: characterising gut microbiome: which subtypes not done? Based on review do authors think need to improve methodologies or sampling? Both?
9. Conclusion: are authors suggest fecal transplantation as a cancer treatment? Would disagree that there is literature to support this… suggest remove or reword. Similarly with prebiotics and probiotics… potentially dangerous/misleading suggestions.
Author Response
The authors have written a short review on the association between colorectal cancer treatment outcomes and microbiome, highlighting the role of the microbiome in immunotherapy.
1. Line 82-85: Expecting authors to talk about immunotherapy use. Suggest restructure paragraph so that immunotherapy comes first.
We thank the reviewer for this comment and have restructured the paragraph according to the suggestion.
2. Line 128 Section 2: title is too vague. Consider renaming section: Role of gut microbiome in modulating immune response to cancer treatments
Thank you we renamed the title of section 2 according to the suggestion.
3. Figure 1: Ambitious and perhaps misleading. Currently diagram suggests that the gut microbiome directly affects response to immunotherapy but this happens via immune system. Described in text as role in development and regulation of adaptive immunity; Suggest remove beneficial effects and dysbiosis and focus on modification of response to therapy via immune system. Write more thorough description to accompany new figure.
Thank you, we have modified the figure as per you suggestion and added text to describe the figure thoroughly.
4. Line 216: reword sentence to simplify
Thank you, the sentence was rephrased.
5. Line 243: the worst? Most aggressive?
Thank you, we removed worst and replaced it with most aggressive.
6. Figure 2: Needs description
Thank you, we added a description of the figure in the legend.
7. List of abbreviations not complete: e.g. dMMR , TGF, CMS1
We thank the reviewer for his comment. dMMR and CMS1 have been added to abbreviation list. However, TGFβ is already present in the list.
8. Conclusion: Authors suggest 3 steps for further research. More detail is needed on each of these. For example: characterising gut microbiome: which subtypes not done? Based on review do authors think need to improve methodologies or sampling? Both?
We thank the reviewer for this comment and have added more detail pertaining to steps for further research.
9. Conclusion: are authors suggest fecal transplantation as a cancer treatment? Would disagree that there is literature to support this… suggest remove or reword. Similarly with prebiotics and probiotics… potentially dangerous/misleading suggestions.
Thank you, we rephrased the concluding sentences in order to deliver the message that microbial therapy in addition to other cancer treatments is a promising therapy but should be taken with caution.